# HeroMaker: Human-centric Video Editing with Motion Priors

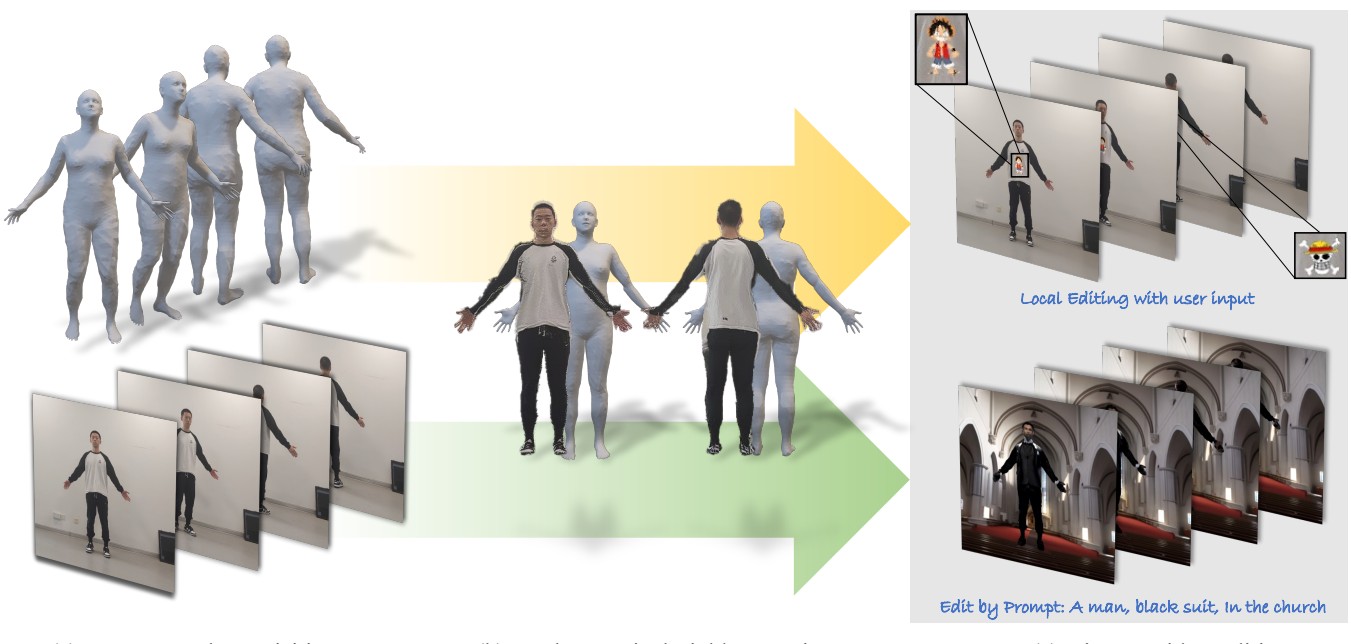

(a) Human Mesh acquisition       (b) Dual Canonical Fields Learning       (c) Diverse Video Editing

**Figure 1: We present HeroMaker, a new video representation with motion priors for human-centric video editing, which contains human motion warping, margin refinements, and dual canonical fields. As illustrated in (a), our model employs the body mesh to portray the structure information of people in the video. From (a) to (b), our model reconstructs the video with explicit human motion warping and neural margin refinements between dual canonical fields and each human-centric video frame. (c) shows the two editing results from HeroMaker, which are temporally consistent and plausible.**

## ABSTRACT

Video generation and editing, particularly human-centric video editing, has seen a surge of interest in its potential to create immersive and dynamic content. A fundamental challenge is ensuring temporal coherence and visual harmony across frames, especially in handling large-scale human motion and maintaining consistency over long sequences. The previous methods, such as diffusion-based video editing, struggle with flickering and length limitations. In contrast, methods employing Video-2D representations grapple with accurately capturing complex structural relationships in large-scale human motion. Simultaneously, some patterns on the human body appear intermittently throughout the video, posing a knotty problem in identifying visual correspondence. To address the above problems, we present HeroMaker. This human-centric video editing framework manipulates the person's appearance within the input video and achieves inter-frame consistent results. Specifically, we propose to learn the motion priors, transformations from dual canonical fields to each video frame, by leveraging the body mesh-based human motion warping and neural deformation-based margin refinement in the video reconstruction framework to ensure the semantic correctness of canonical fields. HeroMaker performs human-centric video editing by manipulating the dual canonical fields and combining them with motion priors to synthesize temporally coherent and visually plausible results. Comprehensive experiments demonstrate that our approach surpasses existing methods regarding temporal consistency, visual quality, and semantic coherence.

ACM MM, 2024, Melbourne, Australia
© 2024 Copyright held by the owner/author(s). Publication rights licensed to ACM.
ACM ISBN 978-x-xxxx-xxxx-x/YY/MM
https://doi.org/10.1145/nnnnnnn.nnnnnnn

## CCS CONCEPTS

• **Computing methodologies → Artificial intelligence**.

## KEYWORDS

Human-centric Video Editing, Diffusion Model, Motion Priors

# 1 INTRODUCTION

Human-centric video editing focuses on modifying the individual within a given video and generating temporally coherent results. This technique has numerous potential applications, such as media content production, virtual reality, and video games. A pivotal challenge in human-centric video editing is maintaining coherence and harmonious results across frames when people can move freely in the video.

Recent diffusion-based video editing explores extracting and incorporating various structural correspondences using infer-frame attention maps [6, 8, 32, 55], optical flows [11, 58] and nn-fields [19]. Although the temporal consistency has improved, it still struggles with flickering and length limitations. Alternatively, some researchers have explored video-2D representations, storing the information of the video in atlases [30] or canonical images [46] to propagate changes over time. However, it grapples with accurately capturing complex structural relationships in large-scale human motion. Moreover, many studies have focused on reconstructing a human body in 3D and attempting to edit it. While promising, these methods often present challenges regarding cost, size, and unfriendly user-controlled environments due to the semantic-less texture maps and the requirement for long-term optimization processes.

Since each human part is unique, patterns on the human body appear intermittently throughout the video due to self-occlusion, posing a knotty problem in identifying correspondence to ensure consistent video editing. As a method of video-2D representation, CoDeF[46] builds correspondence via learning a neural deformation field from a canonical image to each frame and does the video editing on the canonical field, which improves the correspondence between frames. Although it achieves high-fidelity reconstruction, the canonical image differs from natural images, leading to difficulties when editing with image editing tools, including ControlNet [62], and resulting in editing challenges to generate semantically plausible results.

In light of CoDeF's success, we leverage the human body mesh to obtain semantic human canonical images, which provide structural and texture correspondence in 3D space. Our framework defines motion priors, incorporating human motion warping, neural margin refinements, and dual canonical fields to achieve this goal. To obtain the motion priors from a given video, our model employs an off-the-shelf human mesh estimator to set up an initial body mesh. Then, we refine the body mesh in a two-step optimization to close its shape to the person's in videos to ensure more accurate human motion warping. Resorting the motion priors, our model defines the dual canonical fields with a frontal and back body mesh under the A-pose to obtain the vast majority of human body information. Subsequently, it reconstructs the video with explicit human motion warping and neural margin refinements between dual canonical fields and each human-centric video frame. Additionally, our model supports diverse user interactions for modifying the videos. Hero-Maker performs human-centric video editing by manipulating the semantic-aware dual canonical fields. Together with the motion priors, it synthesizes temporally coherent and visually plausible results.

We summarize our contributions as follows:

- We propose a new human-centric video representation combining motion priors and deformation fields to reconstruct and edit the video.
- We leverage the motion priors with human motion warping based on body mesh, neural margin refinements, and dual canonical fields to identify accurate structural correspondence and produce coherent results.
- Extensive experiments demonstrate that our model could produce temporal coherent and plausible results, especially during large-scale human motion.

# 2 RELATED WORK

## 2.1 Text-to-Video Generation and Editing.

Recent works attempt to extend a latent diffusion model into a T2V editing model [2, 5, 6, 15–17, 19, 21, 22, 24, 25, 27, 32, 34, 35, 38, 41, 44, 48, 49, 52, 55, 56, 59–61, 65]. Tune-A-Video [55] and Control-A-Video [8] extend a latent diffusion model to the spatial-temporal domain and finetune it with source videos. However, they still have difficulties in modeling complex motions and long sequences. Text2Video-Zero [32] and ControlVideo [64] use ControlNet [62] to preserve the per-frame structure but struggle to temporal consistency. FateZero [48] and vid2vid-zero [53] use attention maps to enhance shape-aware editing based prompt-to-prompt [23], but they still have temporal issues. Rerender-A-Video [58], TokenFlow [19], and VideoControlNet [27] utilize optical flow to control inter-frame relationships to improve consistency. However, they still face challenges when addressing large-scale human motion and rotation issues. TokenFlow [19] enforces linear combinations between diffusion features based on source correspondences. However, the pre-defined combination weights are not adapted to all videos, resulting in high-frequency flickering. Because TokenFlow [19] needs to cache information for each frame when processing long-sequence videos, resulting in insufficient memory, we will not compare it with this method.

The above methods explore the augmentation of inter-frame attention modeling on a diffusion model. They ensure the correct spatial structure but still challenge temporal consistency. Recently, AnimateDiff [21] presented a motion module trained on extensive video data without a fineturing diffusion model, improving temporal consistency. Furthermore, human-centric videos have further explored some workss [26, 57] and achieved visually plausible results. However, generating the same effect in videos in a small amount of video data poses challenges. Unlike these works, our method leverages human motion priors to achieve text-guided video editing effectively.

## 2.2 Temporal Propagation in video editing.

Another significant line of video editing work relies on a powerful video representation. VideoSnap [66] compresses videos using spatio-temporal feature maps into one or several images and then trains an expansion network to transform these images back into videos. The layered neural atlas [30] factorizes the input video using a layered presentation. It maps the subject/background of all frames using 2D UV maps as an intermediate editing representation. Once the layered neural atlas is learned, editing can occur either on keyframes or on the atlas itself, and the editing results consistently

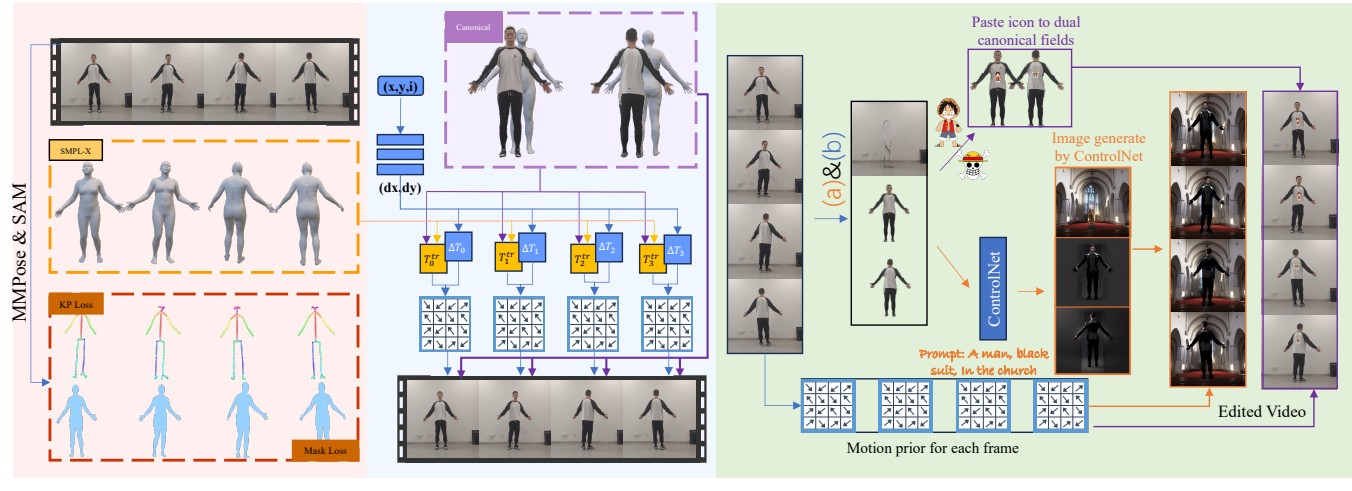

(a) Optimizing SMPL-X     (b) Video Reconstruction     (c) Video Editing

Figure 2: We propose a multi-stage framework for human-centric video editing. We first acquire the motion priors for each frame based on SMPL-X [47] (a). Building upon the motion priors, we devise an editing-friendly video representation to reconstruct the input video (b). Then, our optimized video representation enables superior editing performance as in (c).

propagate to other frames [4, 7, 13, 28, 33]. CoDeF [46] incorporates the 3D deformation field with the 2D hash-based canonical image to improve the video representative capability further. However, Video Snapshot [66], atlas [30], and canonical image [46] all utilize optical flow to help to predict the relationships between each frame. They encounter difficulties in reconstructing and editing videos with large-scale human motion. This leads to incorrect correspondence and texture information, resulting in unnatural results.

## 2.3 3D Human Reconstruction and Editing

3D human reconstruction and editing are closely related to human-centric video editing tasks. Many papers aim to reconstruct an accurate human body and texture using the SMPL+D model [1, 18] or implicit functions [20, 29, 54] through monocular videos. However, most focus on reconstructing and driving more accurate human models without considering editing effects and friendliness. Some works attempt to edit in 3D, SINE [3] and SKED [42] support editing a local region of the base NeRF [43]. Dyn-E [63] and Control4D [51] propose to edit the contents of dynamic NeRFs. However, Dyn-E [63] can only edit the local appearance with user manipulation. Control4D [51] needs multi-view videos as input and can only handle videos with small motions and short video lengths. Recently, DynVideo-E [37] has attempted to edit monocular videos in 3D through text. However, it is not user-friendly for reconstruction and editing operations to take tens of hours to complete. Although these methods can produce high-fidelity results, their cost, size, and controlled environment are unfriendly to users. Instead, video or image editing frameworks are more likely to avoid these shortcomings. Our method introduces the motion priors to the human body. Then, we convert it into pixel position relationship conversion between each frame. While retaining the correctness of the 3D structure, it transforms the task into 2D image editing, which is also one of the primary motivations of our work. Our work focuses on proposing a new human-centric video representation to solve problems with the image or video editing task. Meanwhile, we will not compare these methods. [37, 51, 63]. due to some recent work not being open-source yet.

## 3 METHOD

Given a human-centric video, we aim to modify its visual attributes based on diverse user interactions while maintaining correct structural correspondence and temporal consistency. We tackle this problem with a multi-stage framework, namely reconstructing and editing. As shown in Fig. 2, HeroMaker introduces a novel video representation by leveraging the motion priors based on the SMPL-X [47], which establishes the transformation correspondence from the canonical field to each video frame. In the following, we first illustrate the motion priors in Sec.3.1, and then our novel video representation is elaborated in Sec.3.2, followed by details of the editing procedure and applications of the whole framework in Sec.3.3.

## 3.1 Preliminary: Motion Priors

As previously stated, the visual quality of video editing largely depends on the established video representation. To learn a better video representation for facilitating subsequent video editing, we resort to readily accessible motion priors. Specifically, we mitigate the deformation ambiguity by breaking it down into two components: known human motion warping and neural margin refinement. Thus, our first step is to obtain reliable motion priors.

As depicted in 2 (a), starting with a human-centric video $\{I_i\}_{i=0}^{N-1}$ consists of $N$ frames, we apply the off-the-shelf OSX [36] to predict the camera parameters $P_i$ and SMPL-X [47] coefficients due to its robustness toward partial observations and high efficiency. SMPL-X [47] is defined as a differentiable function $S(\beta, \theta, \psi) \rightarrow (V, F)$ that outputs a 3D human body mesh with 10475 vertices $V \in \mathbb{R}^{10475 \times 3}$, 20908 faces $F \in \mathbb{R}^{20908 \times 3}$, where $\psi \in \mathbb{R}^{10}$ is the facial expression

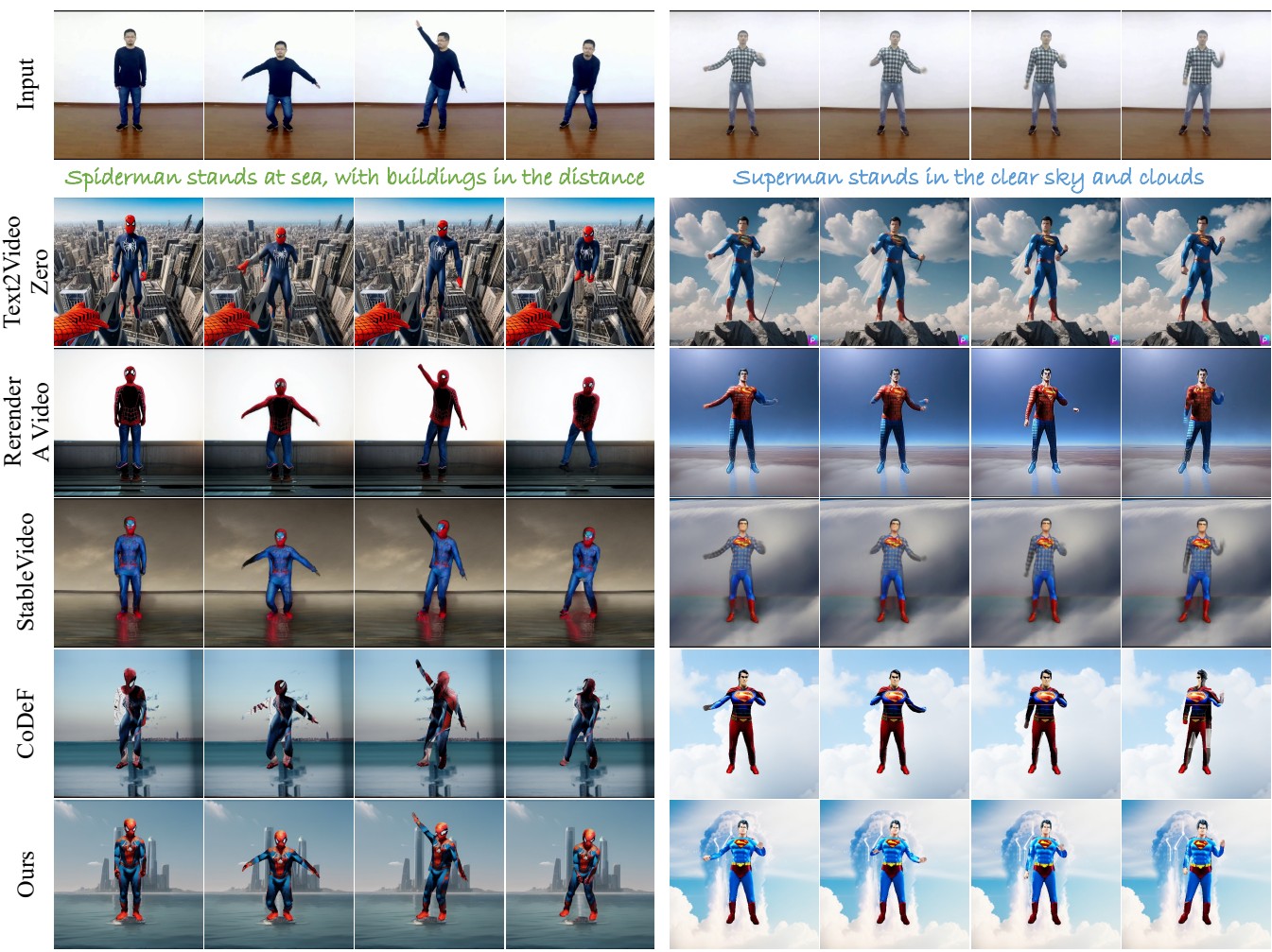

**Figure 3: Qualitative analysis of video editing by the prompt. We compare our method against baselines regarding video editing by a prompt. The first row is the input video, and the colorful description is the corresponding editing prompt. The results indicate that the first three methods suffer from self-occlusion and large-scale movements, thus producing temporal inconsistent results. CoDeF's canonical images differ from the natural ones, leading to results lacking semantics. The results of our method, in the last row, are temporally coherent and plausible.**

parameters, $\beta \in \mathbb{R}^{10}$ and $\theta \in \mathbb{R}^{22 \times 3}$ are the body shape parameters and pose parameters, respectively. In order to enhance the accuracy of the transformation correspondence, instead of directly utilizing the regression-based estimation provided by OSX [36], we adopt a two-step optimization strategy to obtain a more accurate SMPL-X [47] fit.

Firstly, we refine the SMPL-X coefficients with 2d keypoints. Specifically, we leverage mmpose [12] to attain 2D keypoints $P_{i(2D)}$ for each frame $i$. We optimize over the learnable parameters $\theta_{i=0}^{N-1}$ by minimizing the difference between estimated 2D keypoints $P_{i(2D)}$ and corresponding projected 2D joints $P_i(J_{\text{sub}})$, where $P_i$ is the projection matrix. Additionally, we employ a temporal regularization term $\mathcal{L}_{\text{reg}}^1$ on output mesh vertices $V_m^i$ to ensure continuity. The optimization objective of the first stage is:

$$\mathcal{L}^1 = \mathcal{L}_{\text{kps}} + \lambda_{\text{reg}}^1 \mathcal{L}_{\text{reg}}^1 \tag{1}$$

$$\mathcal{L}_{\text{kps}} = \|P_{i(2D)} - P_i(J_{\text{sub}})\|_2^2 \tag{2}$$

$$\mathcal{L}_{\text{reg}}^1 = \|V_m^{[0:n-2]} - V_m^{[1:n-1]}\|_2^2 \tag{3}$$

To further improve the flexibility of the SMPL-X [47] model's expression ability, making it able to match the clothed human better in the video, rather than a skinned person. In the second stage, we added a per-vertex offset $D \in \mathbb{R}^{10475 \times 3}$, to capture the details of each frame and define the model as:

$$S(\beta, \theta, \psi, D) = \text{LBS}(T(\beta, \theta, \psi, D), J(\beta), \theta, W) \tag{4}$$

$$T(\beta, \theta, \psi, D) = T(\beta, \theta, \psi) + D \tag{5}$$

$$T(\beta, \theta, \psi) = T + B_s(\beta) + B_p(\theta) + B_e(\psi) \tag{6}$$

where T is a mean shape template, $B_s$, $B_p$ and $B_e$ are shape, pose and expression blend shapes, respectively. LBS denotes linear blending skinning and $W$ is the vertices' skinning weights. We only optimize $D$ in this stage to avoid overfitting. Since we expect the rendered human outline to align with the foreground mask, we utilize mask loss as a supervision. We use SAM-Track [10] to get a per-frame binary mask $M_i$ of human and minimize the difference between the silhouette of the rendered body $R^s(P_i, \{V_i, F\})$ and the obtained mask $M_i$, where $R^s$ denotes the differentiable silhouette rasterizer. To ensure mesh smoothness, we regulate the offset with Laplacian smoothing loss [14, 45] $L_{\text{Laplacian}}$ and $L_2$ regularization. The optimization objective of the second stage is:

$$\mathcal{L}^2 = \mathcal{L}_{\text{silhouette}} + \lambda_{\text{reg}}^2 \mathcal{L}_{\text{reg}}^2 \qquad (7)$$

$$\mathcal{L}_{\text{silhouette}} = \|R^s(P, \{V_i, F\}) - M_i\|_2^2 \qquad (8)$$

$$\mathcal{L}_{\text{reg}}^2 = L_{\text{Laplacian}}(D) + \gamma\|D\|_2^2 \qquad (9)$$

Finally, we acquire motion priors that is sufficiently expressive for the video.

## 3.2 Video Reconstruction with Motion Priors

We find a powerful video presentation with better temporal continuity than the inter-frame attention model. With the motion priors described in Sec. 3.1, we target a more editing-friendly video representation, which could effectively convert the human-centric video editing problems into image editing problems. While overfitting the observed video using neural representation is relatively accessible, it often leads to a noisy canonical field due to the ill-posed nature of solving the deformation field. Intuitively, the prerequisite for promising editing is establishing a meaningful canonical field. Meanwhile, a well-defined deformation field can relieve the ambiguity in the canonical field, subsequently benefiting high-quality editing outcomes. The previous methods were mainly divided into two types. The first type [30] uses a UV mapping relationship between pixel space and layered neural atlas, which caused inconvenience during editing. The second type [46] wanted to compress video content onto images, but finding the correspondence between frames in large-scale human motion is challenging. Thus, we devise the canonical fields and decompose the temporal deformation in video based on the motion priors. As shown in 2, our video representation comprised of three components:

**Dual canonical fields.** We define the canonical human body as the A-posed SMPL-X+D, $S_c = S(\bar{\beta}, \theta_A, \bar{\psi}) + \bar{D}$ with mean estimated coefficients across video frames. Specifically, we adopt a dual canonical fields design in which we choose the front view $C_{\text{front}}$ and back view $C_{\text{back}}$ of the canonical human body for information complementarity. As for the network structure, our dual canonical fields are constructed using two 2D multi-resolution hash encodings, which map a 2D position $(x, y)$ to $(R, G, B)$ color.

**Human motion warping.** To alleviate the issue of overfitting resulting from directly learning a deformation field [46], we expect that explicit human motion warping dominates the overall deformation, while neural deformation serves as a refinement. Human motion warping is parametric-free, yet it provides semantic correspondences across frames in video representation.

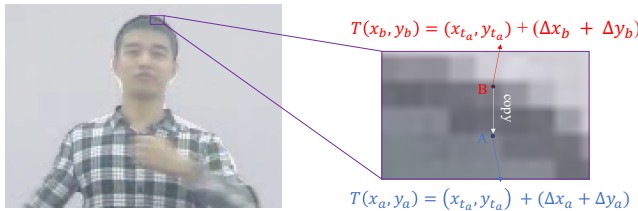

$$T(x_b, y_b) = (x_{t_a}, y_{t_a}) + (\Delta x_b + \Delta y_b)$$

$$T(x_a, y_a) = (x_{t_a}, y_{t_a}) + (\Delta x_a + \Delta y_a)$$

**Figure 4: Margin refine method. Our method deals with the transformation relationships of points outside the mesh transformation matrix region and inside the human mask.**

Inspired by Liquid Warping GAN [39, 40], we build human motion warping using the Neural Mesh Renderer (NMR) [31]. To reconstruct a target frame $I_i$, we first query the canonical fields to acquire two canonical images $I_{\text{front}}$ and $I_{\text{back}}$, and then we embed them into texture space using a weak-perspective camera as $S_c$. Since our motion priors are topologically consistent, we can easily obtain the transformation $T_{\text{front}}^{\text{tr}}$ and $T_{\text{back}}^{\text{tr}}$, which warp the information from the canonical fields to the target frame. Moreover, we compute a mask to fuse the information from $C_{\text{front}}$ and $C_{\text{back}}$. For more details, please refer to the supplementary materials. It is worth noting that we define the deformation in a canonical-to-observation direction, which naturally prevents the drawback of backward deformation [9].

**Neural Margin refinement.** Now, we can make approximate transformations with human motion warping. However, be aware that human motion warping only accounts for rigid transformation, insufficient for clothes and fine-grained non-rigid deformations. In other words, the information in the position $I_i(x, y)$ is not solely determined by the transformation relationship $T_{front|back}^{tr}$. To this end, we design a small refinement field and padding strategy to refine the margin part of the human in a video.

As for the refinement field, we implement 2D multi-resolution hash encoding as the backbone. Specifically, we feed in a triplet $(x, y, i)$ into the refinement field and produce the residual $\Delta T$ : $(\Delta x, \Delta y)$. Formally, the full transformation from front or back-canonical image to each target frame $T = T_{\text{front}|\text{back}}^{\text{tr}} + \Delta T$, where $T_{\text{front}|\text{back}}^{\text{tr}}$ is estimated based on SMPL-X [47].

As shown in Fig. 4, we set the final transformation relationship to be related to the transformation relationship obtained by the human motion warping and margin refinement module.

For points within the transformation relationship $T_{front|back}^{tr}$, we set the full transformation relationship $T_{in}$ as the result of the current point's transformation relationship $T_{front|back(in)}^{tr}$ and add the result $\Delta T_{in}$ refined by the margin refinement module.

$$T_{in} = T_{front|back(in)}^{tr} + \Delta T_{in} \qquad (10)$$

For points outside the transformation relationship $T_{front|back}^{tr}$ and inside the human mask $M$, we set the full transformation relationship $T_{out}$ as the transformation relationship of the point nearest to the effective transformation matrix $T_{front|back(nearest-in)}$, and add the result $\Delta T_{out}$ refined by the margin refinement module.

$$T_{out} = T_{front|back(nearest-in)}^{tr} + \Delta T_{out} \qquad (11)$$

We select the two frames $I_{\text{nn\_front}}, I_{\text{nn\_back}}$ from the video closest to the front-view and back-view canonical field as regularization for training. Specifically, we additionally reconstruct the target frame $\hat{I_i'}$ with the information from $I_{\text{nn\_front}}$ or $I_{\text{nn\_back}}$ according to the orientation similarity of the pelvis. In this way, we can largely preserve the semantic information. Our representation is jointly trained by minimizing the reconstruction loss $L_{\text{rec}}$ between the predicted image $\hat{I}$ and the original one $I$ using mean square error. Moreover, we constrain the output of the deform field using $L_2$ norm empirically. Overall, the loss function of video reconstruction with motion priors can be written as:

$$\mathcal{L}^3 = \mathcal{L}_{\text{rec}}(\hat{I}_i, I_i) + \lambda_{\text{deform}}\|\Delta T\|_2^2 + \lambda_{\text{reg}}^3 \mathcal{L}_{\text{reg}}^3 \qquad (12)$$

$$\mathcal{L}_{\text{reg}}^3 = \sum_{I_i \in \text{front}} L_{\text{rec}}(\hat{I_i'}, I_i) + \sum_{I_i \in \text{back}} L_{\text{rec}}(\hat{I_i'}, I_i) \qquad (13)$$

### 3.3 Video Editing Module

Upon the optimized video representation, we can obtain the trained front canonical image $I_{\text{front}}$ and back canonical image $I_{\text{back}}$ by querying the $C_{\text{front}}$ and $C_{\text{back}}$ with position $(x, y)$. Although our dual canonical fields design retains semantic and structural information, they also introduce a new challenge in terms of semantic consistency. For the randomness during the diffusing and denoising process, editing two canonical images separately using Control-Net [27] may yield inharmonious results. We suggest resolving this issue through a simple yet effective strategy to ensure editing coherence. Specifically, we concatenate the $I_{\text{front}}$ and $I_{\text{back}}$ along the width axis and feed it into ControlNet [27]. The self-attention mechanism implicitly builds the correlation between two canonical images.

As shown in Fig. 2 (c), we explore two different editing scenes: 1) **Video editing by prompt**. Users could modify the content of the human and background separately using text inputs. 2) **Video editing with user input**. Users could directly draw on the canonical images at their will. For example, they can attach a logo to their clothes and automatically propagate it throughout the video.

Moreover, HeroMaker supports editing a person individually within a multi-person video, which differs from most competing methods.

## 4 EXPERIMENTS

### 4.1 Experimental Setup

**Implementation Details.** HeroMaker is implemented in PyTorch. In the first stage, we optimize the mesh with the Adam optimizer($lr = 0.0001$, $\beta = (0.9, 0.99)$) for 200 iterations. The regularization parameter, denoted as $\lambda_{reg}^1 = 0.2$. In the second stage, We optimize for 20 iterations per frame with a learning rate of 0.0005, $\lambda_{reg}^2 = 0.2$ and $\gamma = 10$. During video reconstruction, we jointly trained dual canonical fields and neural margin refinement field together with the Adam optimizer(lr=0.0001, $\beta = (0.9, 0.99)$) for 15000 iterations. We employ the pre-trained Stable Diffusion v1.5 model, and ControlNet [62] provides structure guidance regarding edges. For image editing, we implement 30 timesteps for DDIM sampling.

**Dataset.** We validate the effectiveness of our full pipeline using two datasets, including selected videos from the iPER [39, 40] and

| Method | $E_{\text{vertices}} \downarrow$ | CLIP$\uparrow$ |
|---|---|---|
| Text2Video-Zero [32] | 27.61 | 25.00 |
| Rerender-A-Video [58] | 25.85 | 26.05 |
| StableVideo[7] | 10.53 | 26.43 |
| CoDeF [46] | 26.22 | 27.48 |
| Ours | **7.81** | **27.70** |

**Table 1: Quantitative comparison on prompt-based video editing. We estimate and compute the average mesh vertices error as $E_{\text{vertices}}$ in the original and edited videos. For textual alignment, we report the average CLIP [50] score.**

| Method | Textual fidelity consistency ↑ | Shape preservation↑ | Visual effect↑ |
|---|---|---|---|
| Text2Video-Zero [32] | 0.531 | 0.500 | 0.469 |
| Rerender-A-Video [58] | 0.594 | 0.438 | 0.563 |
| StableVideo[7] | 0.375 | 0.500 | 0.469 |
| CoDeF [46] | 0.375 | 0.344 | 0.344 |
| Ours | **0.813** | **0.625** | **0.625** |

**Table 2: User study on prompt-based video editing.**

| Method | NLA [30] | CoDeF [46] | Ours |
|---|---|---|---|
| Visual effect ↑ | 0.375 | 0.365 | **0.750** |

**Table 3: User study on user interactive video editing.**

in-the-wild internet videos. These videos encompass individuals with diverse body shapes, each performing with different speeds and amplitudes. All videos consist of 50 to 200 frames, and we employ $2 \sim 4$ prompts during editing.

**Baselines.** We compare our method with five baselines: NLA [30], Text2Video-Zero [32], Rerender-A-Video [58], StableVideo [7] and CoDeF [46]. We compare our method with NLA [30] and CoDeF [46] in video editing with the user input task to validate the ability of the model to represent the video in a structure-aware correspondence. For video editing by prompt task, we compare our model with Text2Video-Zero [32], Rerender-A-Video [58], StableVideo [7] and CoDeF [46] to show the temporal consistency and ability to match the prompts.

**Evaluation Metrics.** Human-centric video editing aims to faithfully reflect the editing prompt while maintaining original shape coherency and temporal consistency. We further propose to measure the shape coherency. In detail, we estimate the human mesh in the original and edited video using OSX [36] and compute the average mesh vertices error as $E_{\text{vertices}}$. For textual alignment, we report the average CLIP [50] score, which computes the cosine similarity between the prompts' CLIP [50] embedding and each frame's image embedding in the edited video. However, simply using these metrics cannot fully represent the visual quality of edited videos. We thus conducted a user study. We show the textual descriptions and editing results of different methods, asking them to rate in three aspects: textual fidelity with temporal continuity, shape preservation, and comprehensive visual effect.

## 4.2 Comparison with Baselines

**Quantitative Comparison.** Following previous works, we evaluated our method and baselines with different metrics. As indicated in Table. 1, our method surpasses previous works in all metrics, demonstrating that our editing results align closely with the prompts and maintain the original body shape. We further conduct user studies as described in Sec.4.1. As shown in Table. 2 and Table. 3, the participants exhibit a clear preference for our results.

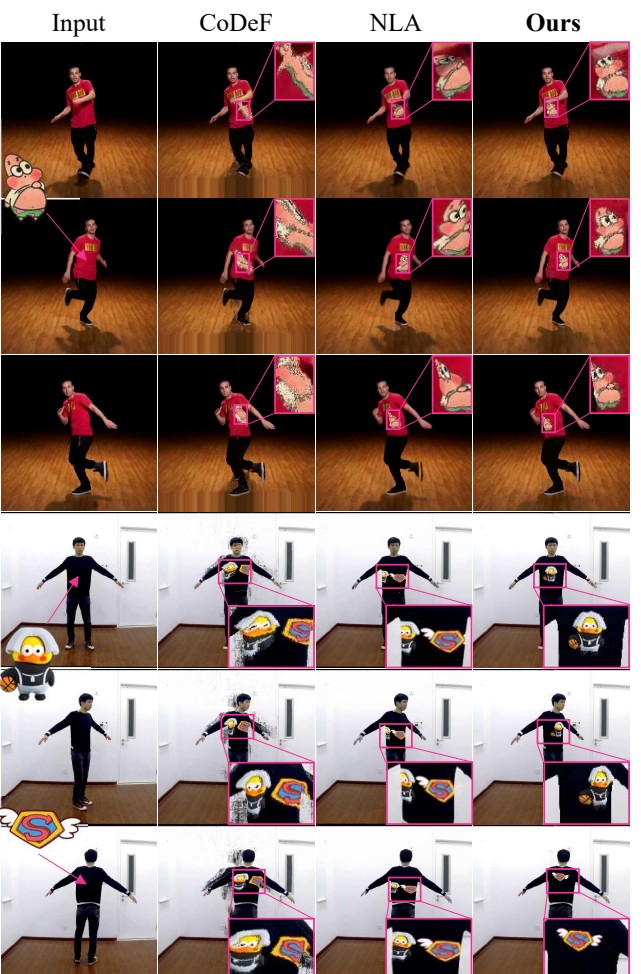

**Figure 5: Qualitative analysis of video editing with user input. Our method supports local editing and allows users to add customized icons accurately into the region of interest. We compare our method against NLA [30] and CoDeF [46].**

**Qualitative Comparison.** Fig.3 presents the visual results of prompt-based video editing. Text2Video-Zero [32] and Rerender-A-Video [58] generate outputs semantically aligned with the text description but fail to maintain temporal consistency. For instance, the body shapes are flickering, and the arms are distorted (see Spider-Man and Superman in Fig. 3). StableVideo [7] exhibits satisfactory temporal consistency. However, it is prone to generating outputs with reduced fidelity. Since CoDeF learns the deformation field and the

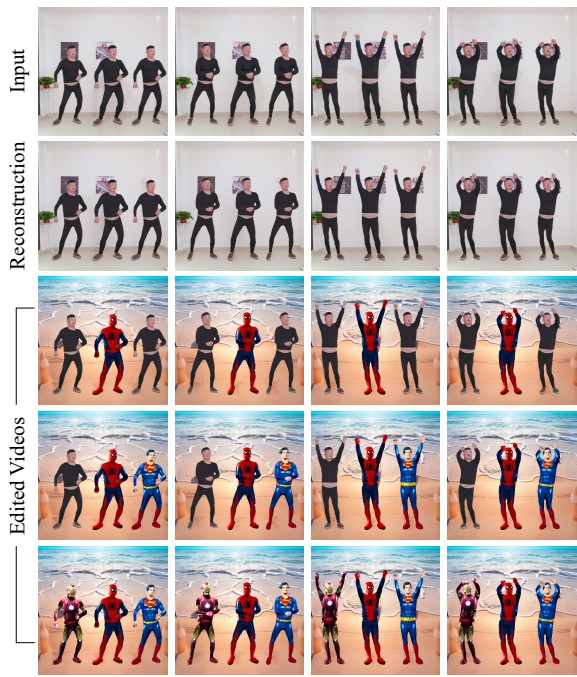

**Figure 6: Multiple people editing results. Our method can extend to the reconstruction and editing, where the input video sequence supports multiple people. In the subsequent stage, users can edit one or more people in the scene individually, providing greater flexibility.**

canonical image without structure information, it generates different results from natural images when handling human-centric videos. According to Fig. 3, by leveraging the motion priors, our method successfully achieves temporal consistency while preserving fidelity.

Furthermore, representing the video with motion priors allows our model to edit the human body locally. It enables users to edit regions of interest while maintaining the other parts. In Fig. 5, we compare our method to NLA [30] and CoDeF [46] in user interactive video editing. NLA [30] and CoDeF [46] models use optical flow to maintain the correspondence between frames. However, estimating optical flow for complex human motion is difficult, which leads to visual flaws. Although NLA [30] shows good textured results, it fails to maintain geometry consistency between human motion. CoDeF [46] leads to information losses when encoding the video into a canonical content field. In some cases, the correspondence of body deformation deviates, causing unpleasant editing results. In contrast, our model utilizes motion priors and thus learns human-aware canonical fields, ensuring that the editing contents are attached to the appropriate positions.

Additionally, unlike most previous methods, HeroMaker offers the ability to easily modify any character within a video containing multiple people, as illustrated in Fig. 6. This capability enhances the appeal and flexibility of human-centric video editing, providing users with a unique and engaging experience.

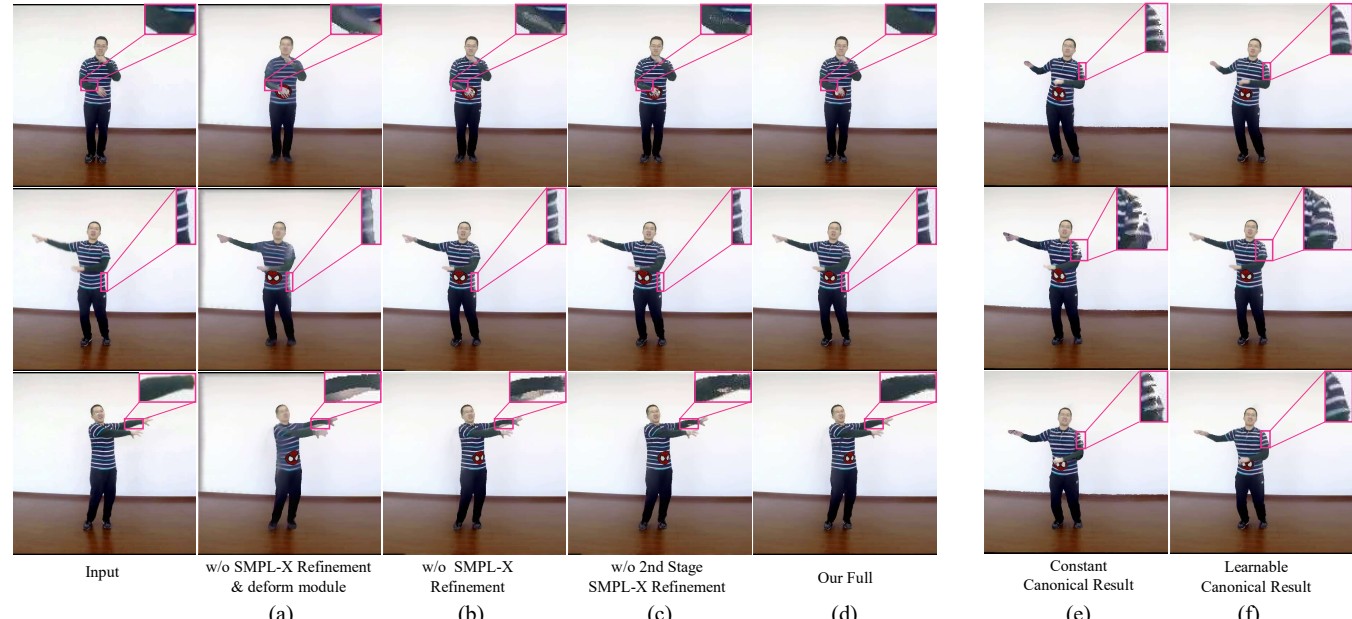

|   | | | | | | |
|---|---|---|---|---|---|---|
| Input | w/o SMPL-X Refinement & deform module | w/o SMPL-X Refinement | w/o 2nd Stage SMPL-X Refinement | Our Full | Constant Canonical Result | Learnable Canonical Result |
|   | (a) | (b) | (c) | (d) | (e) | (f) |

Figure 7: Qualitative ablation results. Compared to method (a), method (b) benefits from a neural deformation module to correct estimation errors from the SMPL-X network. The method (c) further improves the results using first-stage SMPL-X refinement that effectively improves motion priors. Our full uses a neural deformation module and two stages of SMPL-X refinement to achieve precise mesh deformation. Additionally, we demonstrate the necessity of learnable canonical fields by comparing method (e) versus method (f).

## 4.3 Ablation Studies

To verify the contributions of different modules to the overall performance, we systematically deactivate specific modules in our framework and present the visual comparison in Fig.7. In this section, we mainly analyzed the impact of varying degrees of SMPL-X refinement, whether to add a deform module and the necessity of learning canonical fields. We define method (a) as the model without SMPL-X refinement and the neural deformation module. Method (b) incorporates the neural deformation module into the baseline. Considering that our framework performs SMPL-X refinement in two stages, method (c) applies only the first refinement stage, whereas method (d) represents our full model.

**Neural deformation module.** The neural deformation module aims to correct estimation errors from the SMPL-X network by ensuring the alignment of frame images with the canonical fields under motion priors. The comparison between methods (a) and (b) demonstrates improvements in clarity and reduction in edge deviations, highlighting the module's efficacy in enhancing visual quality by learning accurate correspondences.

**SMPL-X refinement.** The refinement process mitigates mesh deviations detected by the SMPL-X network and utilizes 2D keypoints and inter-frame mesh deviations for optimization in the first stage. It is visible that the improvement in image quality in method (c) indicates that the correctness of motion priors has a positive impact on the results. The second refinement stage further rectifies edge artifacts, underscoring the refinement's critical role in achieving precise mesh deformation.

**Learnable canonical fields.** Additionally, to demonstrate the necessity of learnable canonical fields, we select the two images closest to the front and back view as canonical images and then optimize the deform network to obtain the final results. We believe that this can extract as much information as possible from the video while ensuring semantic information. However, as shown in Fig. 7, by comparing method (e) versus method (f), we observe that learnable canonical fields capture more detailed and relevant information from video sequences, thereby reducing reconstruction errors and improving edge definition. Constant canonical images, despite their simplicity, fail to accommodate the complexity and randomness of motion, leading to artifacts in reconstructed images.

## 5 CONCLUSION

In this paper, we present HeroMaker, an innovative human-aware framework that prioritizes human-centric video editing. Our approach utilizes motion priors based on human body mesh to establish the transformation correspondence from human-aware canonical fields to each video frame. Powered by our devised video representation, we maintain meaningful and structural canonical fields that facilitate the subsequent synthesis of temporal coherent and plausible results in response to diverse user interactions. Extensive experiments and visual results demonstrate the superior performance of our HeroMaker, while ablation studies confirm the effectiveness of our design.

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
