# OpenReview forum: "HeroMaker: Human-centric Video Editing with Motion Priors"
_acmmm.org/ACMMM/2024/Conference — MM2024 Oral_

### Official Review · Reviewer_isQA · 2024-05-16

**Rating:** 5
**Confidence:** 2

**Summary:**

In this paper, a human-centric video editing framework named HeroMaker is proposed, which leverages motion priors and dual canonical fields to achieve inter-frame consistent results. By leveraging the motion priors based on the SMPLX, HeroMaker establishes the transformation correspondence from the canonical field to each video frame. A supplemental video demo demonstrates that this approach surpasses existing SOTAs and receives temporally coherent and visually plausible results.

**Strengths:**

1. This paper proposes a novel video representation based on motion priors and breaks it down into two components: known human motion warping and neural margin refinement, which could effectively convert the human-centric video editing problems into image editing problems
2. Benefitting from the introduced neural margin refinement, it is convenient for editing videos of clothed bodies.

**Limitations:**

1. The author claims that by optimizing bias ΔT, this method could solve the video editing problem of clothed human. However, from the pipeline and experiments, clothes generally fit well. What I want to know is, is this method still effective for some loose fitting clothes, like dresses?
2.  The method is light incremental, both motion prior and canonical fields have been presented in previous methods. It is necessary for the author to clarify the innovation and contribution of the paper.
3. For the section related to video reconstruction, especially the part of Neural Margin Refinement and the definitions of 'points within/outside the transformation relationship', I think it is obscure and difficult to understand. Can the author polish this part in the revision?
4. Others:
  -. Line214, finetuning
  -. L216 works
  - L316 Inappropriate references
  -. Paper readability. Too many "however" appear in the section of related work, I suggest that the authors could utilize other similar words.

**Suitability:**

2

---

### Official Review · Reviewer_D4QC · 2024-05-23

**Rating:** 4
**Confidence:** 3

**Summary:**

The paper presents a human-centric framework for video editing, which incorporates human motion priors to establish dual canonical fields, effectively converting video editing tasks into more tractable image editing problems. It demonstrates success in achieving both temporal consistency and plausibility in edited outcomes for videos featuring single individuals and multiple people.

**Strengths:**

1.Important Topic: human-centric video editing is indeed a topic of large importance in the community given the direction of the field.

2.Reasonable 3D Human Optimization Strategy: The paper introduces a fresh perspective by incorporating the optimization of 3D human reconstruction outcomes with a consideration for editing effectiveness and user-friendliness, thereby furnishing video editing tasks with more accurate and suitable human representations.

3.Innovative Deformation Handling: The work cleverly separates deformation into rigid motion and non-rigid margin, and applies a neural refinement technique to the human silhouette edges. This strategy closes the gap between simplified canonical representations and realistic image output, demonstrating an innovative approach to achieving natural-looking edits.

**Limitations:**

1.Video Duration Limitations: The study focuses on videos of up to 200 frames, which may limit generalizability. Including results from longer videos would enhance the validity of temporal consistency claims.

2.Insufficient Differentiation Though the paper cites high computational demands and user unfriendliness in comparison to methods like DynVideo-E, it fails to provide HeroMaker's specific editing times within the Experiments section. Clearer performance comparisons, including timing, are necessary to fully assert the paper's advancements.

3.Static Scene Bias: The framework appears tailored to videos with fixed cameras and stationary subjects. Clarification on or demonstration of HeroMaker's capability to handle videos with moving cameras and dynamic subjects would expand its applicability.

**Suitability:**

2

---

### Official Review · Reviewer_Zdsc · 2024-05-24

**Rating:** 4
**Confidence:** 4

**Summary:**

This paper addresses the challenges in human-centric video editing, focusing on ensuring temporal coherence and visual harmony across frames, particularly when handling large-scale human motion. The authors propose a novel framework, HeroMaker, to tackle these issues by manipulating a person's appearance within input videos to achieve consistent inter-frame results. HeroMaker leverages body mesh-based human motion warping and neural deformation-based margin refinement to ensure semantic correctness in the video reconstruction process. By manipulating dual canonical fields and combining them with motion priors, HeroMaker synthesizes temporally coherent and visually plausible videos.

**Strengths:**

### 1.The paper addresses challenges in human-centric video editing with SMPL approach.
### 2. HeroMaker combines body mesh-based warping and neural deformation-based refinement to ensure temporal coherence and visual quality.
### 3. The user study validate the superior performance of HeroMaker over existing methods.

**Limitations:**

### 1. Image Quality in Figures: The edited results presented in the figures appear somewhat blurred, particularly in terms of texture details. It is unclear whether this is due to issues with the mesh rendering or other factors.

### 2. Comparison with Advanced Methods: The comparison methods primarily include earlier T2V models and NLA-based models, but lack comparisons with more advanced techniques such as Fatezero [1] and Tokenflow [2] (ICLR 2024). From a visual perspective, these newer methods seem to generate finer details and higher quality results. Including a comparison with these state-of-the-art methods would provide a more comprehensive evaluation of the proposed approach’s strengths and weaknesses.

[1] Qi, Chenyang, et al. "Fatezero: Fusing attentions for zero-shot text-based video editing." Proceedings of the IEEE/CVF International Conference on Computer Vision. 2023.

[2] Geyer, Michal, et al. "Tokenflow: Consistent diffusion features for consistent video editing." arXiv preprint arXiv:2307.10373 (2023).

**Suitability:**

3

---

### Official Review · Reviewer_T38u · 2024-05-25

**Rating:** 5
**Confidence:** 2

**Summary:**

The paper targets human-centric video editing which aims to create immersive and dynamic content by modifying human performer within the input video. A significant challenge in this field is maintaining temporal and visual coherency, particularly with large-scale human motion. Prior works struggle with issues like flickering, length limitations, and accurately capturing complex structural relationships.

To overcome these challenges, the authors introduce HeroMaker which utilizes body mesh-based human motion warping (responsible for rigid transformations) and neural deformation-based margin refinement (responsible for non-rigidities) to ensure semantic correctness and temporal consistency. By leveraging dual canonical fields and motion priors, HeroMaker achieves temporally coherent and visually plausible results. The authors present comprehensive experiments to show that HeroMaker outperforms existing methods in terms of temporal consistency, visual quality, and semantic coherence.

**Strengths:**

- The dual canonical that enables identifying more accurate correspondences is novel
- The idea of neural margin refinements is novel in video generation/editing task. It is interesting to adopt commonly used deformation modeling approach in 2D space for video reconstruction.
- The paper presents comprehensive experimental results and comparisons, signifying their superior performance

**Limitations:**

Minor comments:
- L488: presentation -> representation
- L506: in 2 -> Fig. 2

**Suitability:**

2

---

### Meta-Review · Area_Chair_1ko9 · 2024-07-03

**Recommendation:** Accept (Oral)
**Confidence:** 4

**Metareview:**

The paper focuses on human-centric video editing, aiming to create immersive and dynamic content by modifying the human performer within the input video. Their motion representation is novel (though inspired from  existing representations), however, it's significance is underwhelming. The quality of the final results could potentially be improved. A clear limitation is that this method only works with tight-fit clothing, simple poses, and uniform lighting conditions, limiting its applicability to in-the-wild videos. The choice of methods to compare with is not justified properly. Those methods are meant to work on a wide variety of examples. On the other hand, the proposed method is just  tuned for human-centric video. For improved comparisons, the authors can refer to a recent paper (Zhu et al. Champ: Controllable and Consistent Human Image Animation with 3D Parametric Guidance). Please note that I am not holding this against the authors, as this was submitted post the ACM-MM deadline.

I personally feel that the paper is a borderline work. However, considering all reviews and the author response, I am recommending for an accept. Please note that the authors must incorporate the reviewers' suggested changes into the final version of the paper.